# IMPA-Net: Interpretable Multi-Part Attention Network for Trustworthy Brain Tumor Classification from MRI

**DOI:** 10.3390/diagnostics14100997

**Published:** 2024-05-11

**Authors:** Yuting Xie, Fulvio Zaccagna, Leonardo Rundo, Claudia Testa, Ruifeng Zhu, Caterina Tonon, Raffaele Lodi, David Neil Manners

**Affiliations:** 1Department of Biomedical and Neuromotor Sciences, University of Bologna, 40126 Bologna, Italy; yuting.xie2@unibo.it (Y.X.); caterina.tonon@unibo.it (C.T.); raffaele.lodi@unibo.it (R.L.); 2Functional and Molecular Neuroimaging Unit, IRCCS Istituto delle Scienze Neurologiche di Bologna, Bellaria Hospital, 40139 Bologna, Italy; 3Department of Imaging, Cambridge University Hospitals NHS Foundation Trust, Cambridge Biomedical Campus, Cambridge CB2 0SL, UK; fz247@cam.ac.uk; 4Department of Radiology, University of Cambridge, Cambridge CB2 0QQ, UK; 5Department of Information and Electrical Engineering and Applied Mathematics, University of Salerno, 84084 Fisciano, Italy; lrundo@unisa.it; 6INFN Bologna Division, Viale C. Berti Pichat, 6/2, 40127 Bologna, Italy; 7Department of Physics and Astronomy, University of Bologna, 40127 Bologna, Italy; 8Department of Engineering “Enzo Ferrari”, University of Modena and Reggio Emilia, 41125 Modena, Italy; reefing.z@gmail.com; 9Department for Life Quality Studies, University of Bologna, 40126 Bologna, Italy

**Keywords:** decision support, interpretability, trustworthiness, deep neural networks, brain tumor classification, multi-part attention

## Abstract

Deep learning (DL) networks have shown attractive performance in medical image processing tasks such as brain tumor classification. However, they are often criticized as mysterious “black boxes”. The opaqueness of the model and the reasoning process make it difficult for health workers to decide whether to trust the prediction outcomes. In this study, we develop an interpretable multi-part attention network (IMPA-Net) for brain tumor classification to enhance the interpretability and trustworthiness of classification outcomes. The proposed model not only predicts the tumor grade but also provides a global explanation for the model interpretability and a local explanation as justification for the proffered prediction. Global explanation is represented as a group of feature patterns that the model learns to distinguish high-grade glioma (HGG) and low-grade glioma (LGG) classes. Local explanation interprets the reasoning process of an individual prediction by calculating the similarity between the prototypical parts of the image and a group of pre-learned task-related features. Experiments conducted on the BraTS2017 dataset demonstrate that IMPA-Net is a verifiable model for the classification task. A percentage of 86% of feature patterns were assessed by two radiologists to be valid for representing task-relevant medical features. The model shows a classification accuracy of 92.12%, of which 81.17% were evaluated as trustworthy based on local explanations. Our interpretable model is a trustworthy model that can be used for decision aids for glioma classification. Compared with black-box CNNs, it allows health workers and patients to understand the reasoning process and trust the prediction outcomes.

## 1. Introduction

Brain cancer is one of the ten leading causes of death globally among men and women [1,2]. The World Health Organization estimates the 5-year survival rate is only 21% for people aged 40 and over [2]. In most clinical scenarios, LGGs are well-differentiated, slow-growing lesions, while HGGs are usually aggressive with dismal prognosis [3,4]. Survival rates differ markedly for different tumor grades. Identifying tumor grade at an early stage is a major unmet need; it contributes to formulating better treatment strategies and enhances the overall quality of life of patients.

Magnetic resonance (MR) imaging is a non-invasive technique that remains the standard of care for brain tumor diagnosis and treatment planning in clinical practice [5,6]. It provides a reasonably good delineation of the gliomas and conveys biological information on the tumor location, size, necrosis, edema tissue, the mass effect, and breakdown of the blood–brain barrier (which results in contrast enhancement in post-contrast-enhanced T_1_-weighted (ceT_1_w) MR images) [6]. In general, LGGs are less invasive. They usually have well-defined boundaries and homogeneous tumor cores without prominent mitosis, necrosis, and microvascular proliferation [6,7,8,9]. HGGs always show more mass effect. They usually show microscopic peritumoral white matter tract invasion. The demonstration of this diffuse infiltration is an important discriminating feature for the accurate glioma diagnosis [6].

Diagnosis of brain tumors from MR images is a time-consuming and challenging task that requires professional knowledge and careful observation. As alternatives, various automated diagnosis approaches have been developed to assist radiologists in the interpretation of the brain MR images and reduce the likelihood of misdiagnosis. Convolutional neural networks (CNNs) provide a powerful technology for medical data analysis [10]. CNN-based deep learning architectures can extract important low-level and high-level features automatically from the given training dataset of sufficient variety and quality [11]; they embed the phase of feature extraction and classification into a self-learning procedure, allowing fully automatic classification without human interaction, which can be applied to the problem of tumor diagnosis.

Over the last decade, methods using CNNs have been extensively investigated for brain tumor classification due to their outstanding performance with very high accuracy in a research context [12,13]. The differential classification of HGG and LGG is a comparatively simple task that has been tackled in numerous different ways using different CNN methods, and the best-performing models have demonstrated close to 100% performance [10]. For example, Khazaee et al. [14] used a pre-trained EfficientNetB0 for HGG and LGG classification. The model achieved a mean classification accuracy of 98.87%. Chikhalikar et al. [15] proposed a custom CNN model to classify the type of tumor present in MRI images, achieving an accuracy of 99.46%. The authors in [16] used transfer learning with stacking InceptionResNetV2, DenseNet121, MobileNet, Incep-tionV3, Xception, VGG16, and VGG19 for the same classification task. The average classification accuracy for the test dataset reached 98.06%. Zhuge et al. [17] utilized a pre-trained ResNet50. The classification accuracy of the proposed model reached 96.3%.

The above CNN-based methods all achieved remarkable performance on automated HGG and LGG classification. However, MR images are unlikely to be artifact-free [18], and the lesion signal measured by MRI is typically mixed with nuisance sources. The above-mentioned black-box CNNs may learn confounding sources from MR images for decision making, and the health outcomes cannot easily gain the trust of physicians or patients because the evidence is unknown [6,19].

The lack of transparency and interpretability concerning the decision-making process still limits their development into clinical practice [12,19,20]. Visualizing the features that are faithful to the underlying lesion is crucial to ensuring the interpretability and trustworthiness of classification outcomes. Interpretability is the ability to provide explanations in terms understandable to a human [21], based on their domain knowledge related to the task, or common knowledge, according to the task characteristics. The need for interpretability has already been stressed by many papers [21,22,23], emphasizing cases where lack of interpretability may be harmful. Can we explain why algorithms go wrong? When things go well, do we know why and how to exploit them further?

In order to deploy a system in practice, it is necessary to present classification results in such a way that they are acceptable to end users. This is only possible if users trust the decision-making process, which, as a consequence, must be transparent and interpretable. To date, a limited number of saliency-based interpretable methods have suggested different frameworks to improve the interpretability and trustworthiness of CNNs for brain tumor classifications [24,25,26,27]. We divide the previous interpretable approaches into two categories: object-level methods and pixel-level/part-level methods.

At the coarsest level, there are models that have been proposed to offer object-level explanations for brain tumor classification tasks, such as a class activation mapping method GradCAM [24,25] that highlights that entire object as the explanation behind the tumor predictions. The authors in [25] proposed a pre-trained ResNet-50 CNN architecture to classify three posterior fossa tumors and explained the classification decision by using GradCAM. The heatmap generated by the GradCAM technique can identify the area of emphasis and help visualize where the classification model looks for individual predictions.

At a finer level, there are a few interpretable techniques that have been applied to explain the brain tumor classification results with pixel-level/part-level explanations, such as pixel-level interpretable algorithms SHAP, Guided Backpropagation (GBP) [24], and a part-level interpretable model called LIME. Authors in [27] explained the tumor predictions made by the CNN model with SHAP and LIME methods. The SHAP algorithm explains the individual prediction by computing the contribution of each pixel on a predicted image to the prediction using Shapley values to understand what are the main pixels that affect the output of the model [28]. The LIME algorithm is a counterfactual explanation method that approximates the classification behavior of a complex neural network using a simpler, more understandable model without exploring the model itself [29]. In the study, the authors segmented the input image into superpixels and made small disturbances around each superpixel to figure out the contribution/importance of each superpixel to the prediction result. Another study conducted by Pereira et al. [24] utilized GradCAM and GBP maps to provide insights into the regions that support the prediction to perform quality assessment of tumor grade prediction between HGG and LGG. The GBP is a gradient-based visualization method that can visualize which pixels in the input image are more informative for the correct classification.

The above methods identify the most important pixels or objects of an image as the explanation for the prediction outcomes. To some extent, they verify the validity of the classification models. Nevertheless, it is worth stressing that knowing the most important pixels or objects of an image that determined a specific prediction does not always amount to a good-quality explanation.

Ideally, networks should be able to explain the reasoning process behind each individual decision, and this process, ideally, would be similar to that used by a radiologist, who looks at specific features of the MR image relevant to the task. For example, if a doctor classifies a tumor as HGG, this decision always relies mainly on the high-level class-representative features or properties, like the tumor’s irregularity, the necrotic area, or the enhancing ring [30].

The objectives of this study were to build an interpretable multi-part attention [31] network (IMPA-Net) for brain tumor classification to unbox the model and the reasoning process of individual predictions with understandable MR imaging features. The proposed IMPA-Net, motivated by [32], provides both global and local explanations for brain tumor classification on MRI images. Figure 1 gives a more detailed illustration of the connections and distinctions between the two explanations. The global explanation is represented by a group of feature patterns that the model learns and uses for the classification. The quality of the feature patterns can be used to evaluate the ability and reliability of the model on the classification task. The local explanation interprets the reasoning process of an individual prediction by comparing the prototypical parts of the image with feature patterns. It can be used to evaluate the trustworthiness of individual predictions.

The main contribution of this paper is that it addresses the black-box problems of CNN classification models for glioma diagnosis by developing a model with the following characteristics:(i)The first multi-part interpretable model that can provide both global and local explanations for brain tumor classification, enabling better human–machine collaboration for decision aid.(ii)It presents the reasoning process of individual predictions to show how the model arrives at the decision making in this context, allowing health workers to evaluate the reliability of the prediction outcomes.(iii)It allows the prediction results to be interpreted in a clinical context.(iv)It highlights the most relevant information for predictions based on medical disease-related features that can be understood and interpreted by clinicians and patients.

The remainder of the paper is structured as follows. Section 3 gives a detailed introduction to the dataset, the proposed interpretable multi-part attention network, and the experimental setup. Results are given in Section 3. Section 4 evaluates the performance of the proposed method on both aspects of its classification and explanation. Section 5 concludes the key findings of this study. Section 6 concludes the proposed work and discusses the future research directions.

## 2. Materials and Methods

The overall workflow of the development and evaluation of the proposed methodology is shown in Figure 2. Input brain MRI images are firstly pre-processed by resizing, normalization, and cropping, and then three augmentation methods, including rotation, shearing, and skewing are performed to produce the training dataset. The proposed methodology classifies the input image by comparing its prototypical patches with pre-learned feature patterns of classes HGG and LGG. In this stage, feature patterns of both classes are optimized and produced. The quality of the feature patterns is evaluated in the next step on aspects of their interpretability, class representability, and correctness, and then poor-quality feature patterns are excluded in the local explanation process. In the next stage, local explanations of individual predictions are given to illustrate how the model arrives at the final decisions, and each case will be evaluated based on whether it satisfies two basic conditions identified for reliability assessment. Finally, the proposed model is evaluated on both aspects of its performance (classification and explanation), including classifier performance, global explanation evaluation, local explanation evaluation (correctness and confidence), and user evaluation.

### 2.1. Data and Image Processing

We trained and evaluated our network on data from the BraTS 2017 database [33,34,35]. The dataset contains 285 routine-acquired 3T multimodal clinical MRI scans from multiple institutions, comprising 210 patients with pathologically confirmed HGG and 75 patients with LGG. All images from the dataset were pre-processed by co-registration to the same anatomical template, interpolation to the same resolution (1 mm^3^), and skull stripping [33].

Slices that contain gliomas were extracted from each patient’s MRI scan. Considering the enhancing ring in post-contrast-enhanced T_1_-weighted (ceT_1_w) MR is an important discriminating feature for accurate tumor diagnosis between HGG and LGG [6], in our experiments, only ceT_1_w MR images were considered. The dataset was then partitioned into a training dataset (70%) and a testing dataset (30%). A push dataset of 60 images was randomly selected from the training dataset (30 images for each class).

All images were normalized by Z-score normalization and converted to PNG format, and then the background pixels were cropped to focus feature learning on the brain areas instead of the whole image. Moreover, the images were resized to 224 × 224 to fit the model’s training configurations.

### 2.2. Data Augmentation

To increase the size and variability of the training dataset, data augmentation methods were performed, including twice rotating in the axial imaging plane by a random amount between 20° left and 20° right, shearing by a random amount between 10° left and right twice in the transverse direction, and skewing by tilting the images left/right by a random amount (magnitude = 0.2) twice. In this way, the training dataset is augmented six-fold, resulting in 6228 images (3546 HGG, 2682 LGG).

### 2.3. Interpretable Convolutional Neural Network

Figure 3 gives an overview of the proposed IMPA-Net, which consists of a feature extractor, multi-part attention (MPA), and similarity-based classifier. Images are first propagated into convolutional layers for feature extraction, with a structure selected from VGG16. In the proposed classification model, we chose VGG16 as the feature extractor as it combines simplicity, ease of implementation, and fine-tuning capability with adequate feature extraction effectiveness and generalization ability. The pre-trained VGG16 model is suitable for transfer learning or fine-tuning as a feature extractor for brain tumor classification tasks [12]. A non-linear activation function ReLU is used for all convolutional layers. Then, these convolutional layers are followed by a multi-part attention module for similarity calculation between CNN outputs and the feature patterns pre-learned by the model. In particular, our network tries to find evidence for an image (such as the pre-processed HGG image in Figure 3) to be of class HGG by comparing its prototypical patches with learned feature patterns of class HGG and LGG, as illustrated in the similarity correlation units. This comparison produces a map of similarity scores of each feature pattern, which is upsampled and superimposed on the input image to see which part of the input image is activated by each feature pattern. The activation maps are then propagated into a max-pooling layer, producing a single similarity score for each comparison. Finally, the model classifies the input image based on the top 10 similarity scores. The output $S_{c_{HGG}}$ denotes the weighted sum of top-10 similarity scores generated by the multi-part attention module.

#### 2.3.1. Feature Extractor

The architecture consists of a regular convolutional neural network for feature extraction with a structure selected from VGG16 (kernel size 3×3), followed by two additional 1 × 1 convolutional layers. All these convolutional layers (f) use a ReLU with a non-linear activation function.

For a given pre-processed input image x (such as the HGG sample image in Figure 3), the convolutional layers f extract useful features from x to use for prediction, whose output $c_{out}=f(x)$ have spatial dimension D×7×7, where D is the number of the output channels of the last convolutional layer.

#### 2.3.2. Multi-Part Attention

In our experiments, we allocated a pre-determined number of feature patterns $FP={\left\{{fp}_{i}^{c_{j}}\right\}}_{i=1}^{m}$, m=50 for each class, where $c_{j}$ (j∈{HGG,LGG}) represents the class identity of the feature pattern and i is the index of that feature pattern among all feature patterns of class $c_{j}$. So that for each class, 50 feature patterns are learned and produced by the model from a push dataset. This dataset consists of a pre-determined number of MRI images that are randomly selected from the training dataset. The shape of each pattern is D×h×w, where h×w<7×7. In our experiments, h and w are set to 1. The depth of each feature pattern is the same as that of $c_{out}$ but the height and width are smaller than those of the $c_{out}$, each feature pattern will be supposed to represent some representative activation pattern in a patch of the convolutional output $c_{out}$, which in turn will correspond to some prototypical image patch in the original training image.

In our network, every feature patch can be considered as a representative pattern of one image from the push dataset, and these feature patterns are supposed to direct attention to enough medical semantic content for recognizing a class [36]. As a schematic illustration of the multi-part attention for the HGG sample image in Figure 3, the first feature pattern ${fp}_{1}^{c_{HGG}}$ corresponds to the necrotic tumor core of an HGG training image, and the fourth feature pattern ${fp}_{4}^{c_{HGG}}$ enhancing tumor margin of an HGG training image, and the ninth feature pattern ${fp}_{9}^{c_{HGG}}$ the edematous area of an HGG image.

The similarity correlation units SCU in a multi-part attention module computes the L2 distance between the CNN outputs and the feature patterns, as shown in Equation (1). The $i_{th}$ similarity correlation unit ${SCU}_{i}^{c_{j}}$ of class ${\;c}_{j}$ calculates the squared Euclidean distances between feature patterns ${fp}_{i}^{c_{j}}$ and each patch $\tilde{c_{out}}$ generated from the convolutional outputs $c_{out}$ and then inverts the distances to similarity scores. Mathematically, the similarity correlation unit ${SCU}_{i}^{c_{j}}$ calculates the following:(1)$$dist\left(\tilde{c_{out}},\;{fp}_{i}^{c_{j}}\right)={\left\|\tilde{c_{out}},{fp}_{i}^{c_{j}}\right\|}_{2},\;\tilde{\;c_{out}}\in patches\left(c_{out}\right),$$
(2)$$sim\left(\tilde{c_{out}},\;{fp}_{i}^{c_{j}}\right)=\log\left(\frac{{dist\left(\tilde{c_{out}},\;{fp}_{i}^{c_{j}}\right)}^{2}+1}{{dist\left(\tilde{c_{out}},\;{fp}_{i}^{c_{j}}\right)}^{2}+\varepsilon }\right),$$
(3)$${SCU}_{i}^{c_{j}}\left(c_{out}\right)=\begin{array}{c} max\\ \tilde{c_{out}}\in patches\;\left(c_{out}\right) \end{array}sim\left(\tilde{c_{out}},\;{fp}_{i}^{c_{j}}\right),$$

These similarity scores calculated by Equation (2) define an activation map, which retains the spatial relation of the convolutional output $c_{out}$. The activation map can be unsampled to the size of the input image to visualize the part of the input image that looks most similar to the feature pattern [36]. In Figure 3, the similarity score between the first feature patterns ${fp}_{1}^{c_{HGG}}$, a an HGG necrotic tumor core, and the most activated patch of the input image of a an HGG is $s_{1}^{c_{HGG}}$. The similarity score between the fourth feature pattern ${fp}_{4}^{c_{HGG}}$, an HGG enhancing tumor margin, and the most activated patch of the input image is $s_{4}^{c_{HGG}}$. The third feature pattern ${fp}_{9}^{c_{HGG}}$, an HGG edematous area, activated mostly on the edematous tissue of the HGG sample image, with a similarity score of $s_{9}^{c_{HGG}}$. This shows that our model finds that the necrotic tumor core of the HGG sample image has a stronger presence than that of enhancing tumor margin in the input image.

Equation (2) indicates that the similarity is monotonically decreasing with respect to the squared Euclidean distance, that is, the highest similarity score of the similarity correlation unit ${SCU}_{i}^{c_{j}}$ comes when $\tilde{c_{out}}$ is the closest patch to ${fp}_{i}^{c_{j}}$. In activation maps, warmer values indicate higher similarity between the learned feature patterns and the parts of the input image activated by the feature pattern, which is enclosed in the yellow rectangles on the superimposed source images. Then, the activation maps produced by similarity scores are max pooled to reduce to a single similarity score $s_{i}^{c_{j}}$ for each feature pattern ${fp}_{i}^{c_{j}}$. Hence, if the similarity score of the $i_{th}$ similarity correlation unit ${SCU}_{i}^{c_{j}}$ is high, it indicates that there is a patch in the input image that is very similar to the $i_{th}$ feature pattern of class ${\;c}_{j}$ in the latent space, and that the activated patch contains a similar pattern to that represented in the $i_{th}$ feature pattern.

#### 2.3.3. Similarity-Based Classifier

Finally, in the classifier block, the top 10 ranking similarity scores are multiplied by the class-connection weight matrix $\omega_{i}^{c_{j}}$ to produce the output logit to class $c_{j}$. The matrix $\omega_{i}^{c_{j}}$ represents the relationship between feature patterns and the logit of the class. Higher class-connection values refer to higher representability of the feature pattern to its class.
(4)$$S_{c_{j}}=\sum_{i=1}^{10}\omega_{i}^{c_{j}}\cdot s_{i}^{c_{j}},j\in \left\{HGG,LGG\right\}$$

### 2.4. Model Training

The training of the proposed model is divided into three stages: stochastic gradient descent (SGD) of layers before the classifier layer, projection and optimization of feature patterns, and optimization of class-connection weights.

#### 2.4.1. Stochastic Gradient Descent (SGD) of Layers before the Classifier Layer

The architecture aims to learn meaningful and teak-relevant features that can be used to distinguish between HGG and LGG, where the most important patches for the classification task are clustered (in Euclidean distance) around similar feature patterns of the ‘correct’ class and separated from feature patterns from a different class [36]. To learn these features, an iterative algorithm SGD is used to simultaneously optimize the parameters of the convolutional layers f ($f_{conv}$) in the feature extractor and the feature pattern $FP={\left\{{fp}_{i}^{c_{j}}\right\}}_{i=1}^{m}$ in the multi-part attention module via back propagation. In this step, the weight matrix (class connection values) $\omega_{i}^{c_{j}}$ of the last layer in the classifier block is frozen.

Formally, let $X=\{x_{1},x_{2},\ldots ,\;x_{n}\}$ be a set of training images, $Y=\{y_{1},y_{2},\ldots ,y_{n}\}$ be the set of the corresponding labels. The optimization problem to be solved here is to minimize the defined loss function that incorporates the cross-entropy loss (CELoss), cluster loss (ClstLoss), and separation loss (SepLoss):(5)$$Loss=\frac{1}{n}\sum_{k=1}^{n}CELoss\left(f\circ SCU\circ f\left(x_{k}\right),y_{k}\right)+r_{1}ClstLoss+r_{2}SepLoss$$
where ClstLoss and SepLoss are
(6)$$ClstLoss=\frac{1}{n}\sum_{k=1}^{n}\begin{array}{c} argmin\\ i:{fp}_{i}^{c_{j}}\in {FP}^{y_{k}} \end{array}\begin{array}{c} argmin\\ \;\tilde{f(x_{k})}\in patches(f(x_{k})) \end{array}{\left\|\tilde{f(x_{k})}-{fp}_{i}^{c_{j}}\right\|}_{2}^{2}$$
(7)$$SepLoss=-\frac{1}{n}\sum_{k=1}^{n}\begin{array}{c} argmin\\ i:{fp}_{i}^{c_{j}}\notin {FP}^{y_{k}}\; \end{array}\begin{array}{c} argmin\\ \tilde{f(x_{k})}\in patches\left(f\left(x_{k}\right)\right) \end{array}{\left\|\tilde{f(x_{k})}-{fp}_{i}^{c_{j}}\right\|}_{2}^{2},$$

The CELoss penalizes misclassification during the training process, and the aim is to minimize CELoss to give better classifications. The ClstLoss is minimized to encourage the prototypical parts to cluster around the correct class, see Equation (6), whereas the SepLoss is minimized to separate the prototypical parts from the incorrect class; see Equation (7).

#### 2.4.2. Projection of Feature Patterns

To visualize which parts of the training images from the push dataset are used as feature patterns, the network projects every feature pattern ${fp}_{i}^{c_{j}}$ onto the closest patch of the output $f({x_{k}}^{c_{j}})$ that has the smallest distance from ${fp}_{i}^{c_{j}}$, and the closest patch has the same class ${\;c}_{j}$ as that of ${fp}_{i}^{c_{j}}$ [32]. The reason is that the patch of training image ${x_{k}}^{c_{j}}$ that corresponds to ${fp}_{i}^{c_{j}}$ should be the one that ${fp}_{i}^{c_{j}}$ activates most strongly on. We can visualize the part of ${x_{k}}^{c_{j}}$ on which ${fp}_{i}^{c_{j}}$ has the strongest activation by forwarding ${x_{k}}^{c_{j}}$ through a trained network. Mathematically, for feature pattern ${fp}_{i}^{c_{j}}$ of class ${\;c}_{j}\;$(j∈{HGG,LGG}), the network performs the following update:(8)$$fp_{i}^{c_{j}}=\underset{\mathrm{patch},\mathrm{patch}\in \mathrm{patches}\left(f\left(x^{c_{j}}\right)\right)}{argmin}∥\mathrm{patch}-fp_{i}^{c_{j}}{∥}_{2},y_{k}=c_{j}$$

#### 2.4.3. Optimization of Class-Connection Weights

In this stage, all the parameters from the convolutional layers and multi-part attention blocks are frozen, and a convex optimization on the class-connection weight matrix $\omega_{i}^{c_{j}}$ of the last layer is performed. To rely only on positive connections between feature patterns and logits, the negative connection $\omega_{i}^{c_{j}}$ is set to 0 for all to reduce the reliance of the model on a negative reasoning process of the form “this image is of class HGG because it is not of class LGG.”. Mathematically, we perform this step to optimize
(9)$$\underset{\omega_{i}^{c_{j}}}{\min}\frac{1}{n}\sum_{k=1}^{n}CELoss\left(f\circ SCU\circ f\left(x_{k}\right),y_{k}\right)+\lambda \sum_{c_{j}:{fp}_{i}^{c_{j}}\notin {FP}^{y_{k}}}\left|\omega_{i}^{\left(k,c_{j}\right)}\right|,$$

### 2.5. Experimental Setup

All the experiments were conducted on a PC with an Intel Core i7-6700K 4.00 GHz processor running Ubuntu 18.04.6 with one NVIDIA GeForce RTX 2060, using Python 3.9.7 and PyTorch 1.10.1.

The parameters of the convolutional layers from the VGG16 model were pre-trained on ImageNet [37], and the parameters of the additional convolutional layers were initialized with Kaiming uniform methods [38]. The parameters of the two additional convolutional layers are trained and optimized with the learning rate $3\times 10^{-3}$ for 5 epochs, while the pre-trained parameters and biases are fixed. In the following joint training stage, the parameters of all convolutional layers are optimized from epoch 6, and the model performs feature pattern projection every 20 epochs, that is, epochs 20, 40, 60, 80, and 100, and the convex optimization of the last layer is performed after each feature pattern projection process for 20 iterations with learning rate $10^{-4}$.

The other hyperparameters are learning rate for layers pre-trained on ImageNet: $10^{-4}$ and learning rate for feature pattern optimization: $3\times 10^{-3}$. For VGG16, we set D=128 as the number of channels in a similarity correlation unit.

## 3. Results

### 3.1. Global Explanation

Global explanation can be interpreted as the class-representative features the entire model uses to distinguish two classes. Figure 4 shows six learned feature patterns and their activation maps for each class. It can be seen that all feature patterns localize important distinguishing features of both classes. The feature patterns of HGG that have higher responses in contrast-enhancing tumors as a classification feature agrees with the actual imaging characteristics of HGG [6]; the feature patterns that focus on the necrotic tumor core that present heterogeneous high signal and the edematous areas are also important disease-representative features of HGG [6]; the feature patterns of LGG present higher responses on the homogeneous tumor cores and the non-enhancing tumor margins [7,8,9]. It is worth mentioning that those localized medical features can be understood and interpreted by the users, and thus, our framework can help provide global explanations in a human-understandable manner.

### 3.2. Local Explanation: Individual Predictions

The local explanation of individual predictions has to satisfy two conditions in order for its prediction explanations to be considered trustworthy and reliable; that is, all feature patterns that present the 10 highest similarity scores are from the class of the test image, and the concept of each top-10 feature pattern is consistent with that of the activated prototypical patch.

Figure 5 shows the reasoning process of our interpretable model in reaching a prediction on a test image of an HGG. As shown in the activation maps, the highest responses were found on the tumor core activated by the top and 2nd ranked feature patterns of class HGG (with similarity scores 8.143 and 8.105, respectively), the 3rd ranked feature pattern on the tumor enhancing margins, the 6th, 8th, and 9th ranked feature patterns on the edematous tissues.

The network correctly classifies the tumor as an HGG according to the ground truth. Furthermore, it provides the evidence of this prediction outcome with multi-part attention between patches of the test image and feature patterns as the tumor is classified as an HGG because prototypical patches of the test image, including its necrotic tumor core, enhancing margins, and edematous tissue was found to have higher similarity (top 10) with feature patterns from HGG class. The evidence is evaluated to be trustworthy according to the two reliability criteria, that is, all top-10 feature patterns are from the HGG class, and the concept of each top-10 feature pattern is consistent with that of the localized prototypical patch.

Figure 6 shows the reasoning process for reaching a classification decision on a test image of an LGG. As shown in the third column, the highest responses were found on the tumor core of the LGG image activated by two ‘tumor core’ feature patterns (similarity score of 7.420 and 7.332, respectively), the 3rd and 4th ranked feature patterns on the tumor margins. The network correctly classifies the tumor as an LGG. The explanation is the network classifies the tumor as an LGG because prototypical patches of the test image, including its homogeneous tumor core and non-enhancing tumor margins, were found to have higher similarity (top 10) with feature patterns from the LGG class. Those medical feature patterns can be understood and interpreted by the users, and thus, our framework can help provide global explanations in a human-understandable manner. The evidence for the prediction is evaluated to be trustworthy according to the two reliability criteria.

## 4. Performance Evaluation

### 4.1. Classification Performance

Statistic metrics for classification performance, including accuracy (ACC), precision (PRE), specificity (SPE), sensitivity (SEN), and F_1_-score, were calculated both for the interpretable decision-aid system described in this work and the baseline model, whose architecture consisted of the same convolutional layers without the intermediate multi-part attention module and similarity-based classifier. Correct predictions were further evaluated on their reliability based on local explanations to obtain reliable prediction accuracy to assess the trustworthiness of the model.

Table 1 presents the comparison of the classification performance of our interpretable model (before and after the exclusion of ‘background’ feature patterns) with the baseline model trained on the same dataset. Results show that the interpretable model is slightly less accurate than the baseline model and that the exclusion of the ‘background’ feature patterns improved the classification accuracy by 6.53%.

### 4.2. Explanation Performance

#### 4.2.1. Global Explanation Evaluation

Once trained, the system provides global explanations in the form of a set of feature patterns that identify image features characteristic of the classes to be predicted. Each of the feature patterns learned by the system was evaluated on whether it corresponds to a feature of the class (HGG or LGG) that it is supposed to represent and whether the area with the highest response (red) is located within the tumor or tissue altered by the presence of the tumor. A feature pattern is considered invalid if its most activated area is situated in the background regions, namely healthy tissue, ventricles, non-brain tissue, or image background.

Within all feature patterns, two apparent duplicates were found of the LGG class. Thirteen invalid ‘background’ feature patterns (6 HGG and 7 LGG) were found to have higher responses in regions irrelevant to the classification task (e.g., low-signal ventricles and high-intensity background areas). The accuracy of global explanation, defined as the fraction of learned feature patterns that focus on task-relevant regions, was 86%. The initial assessment process was conducted by one author (Y.T.X). In cases of ambiguity, feature patterns were reviewed by other authors (F.Z, L.R), and the final evaluation was arrived at by consensus. Considering the impact of invalid feature patterns on local explanation, those ‘background’ feature patterns were excluded in the further local analysis process.

Figure 7 evaluates the representability of two feature patterns that have the largest class connection weight of each class. The similarity score between the feature pattern (class connection of 0.737 to HGG) and the prototypical patch from the tumor core of the first HGG sample image ranks #2 with a similarity of 8.020 (max. 8.782) and #4 with a similarity of 7.653 (max. 8.379) with the prototypical patch from the tumor core of the second sample image, showing its high representativity of class HGG. The feature pattern of LGG with the highest class-connection value (1.311) also shows high representativity of class LGG; the similarity scores with the localized patches rank first among 10 feature patterns for two LGG sample images.

#### 4.2.2. Local Explanation Evaluation

The reliability of an individual prediction was evaluated based on whether its local explanation satisfies two basic reliability conditions, namely that the reasoning process should be both confident and correct.

*Confidence in the reasoning process*. Confidence in the reasoning process can be evaluated by examining the output of the local explanation. For each case in the test set, the number of feature patterns corresponding to a correctly or incorrectly identified tumor type was counted among those with the 10 highest similarity scores. The results were averaged and summarized in Table 2. Results demonstrate that the inconsistency of the feature patterns with the class of test images has a high impact on the classification performance of the model (comparison between wrong predictions and correct predictions, mean 8.62, 0.34, respectively) and a small impact on the reliability of the predictions (comparison between correct predictions and unreliable predictions).

*Correctness of the reasoning process.* A correct reasoning process is defined as one in which the concept of the activated prototypical patch is consistent with that of the feature pattern. Table 3 summarizes the number of incorrectly activated background patches by top 10 feature patterns among all test images, unreliable predictions, wrong predictions, and correct predictions.

Wilcoxon Signed-Ranks tests were used to assess the effect of incorrectly activated background patches on the number of mismatched feature patterns among the top 10 ranked, an indicator of prediction reliability, comparing image background areas and brain background (i.e., healthy tissue or CSF), for each classification class both separately and jointly.

Considering all test images, image background showed a significantly lower influence (*p*-value < 0.05) on reliability compared to brain background (W = 1244.5, *p*-value < 0.001), and the same pattern was repeated considering only the HGG test images (W = 411.0, *p*-value = 0.002) or the LGG test images (W = 214.0, *p*-value = 0.005). Dividing the test images based on correct predictions (reliable predictions and unreliable predictions, HGG: W = 171.5, *p*-value = 0.011; LGG: W = 114.5, *p*-value = 0.003), wrong predictions (HGG: W = 53.5, *p*-value = 0.095 (*p* > 0.05), LGG: W = 17.5, *p*-value = 0.943 (*p* > 0.05)) and unreliable predictions (same values as correct predictions), only in the second group did these two sources of error show no difference in their effect.

Table 2 and Table 3 indicate the necessity and importance of unboxing the inference process of CNN models for brain tumor classification. This allows health workers to screen out unreliable ‘correct’ predictions that might have been learned from irrelevant regions for decision making.

## 5. Discussion

This work proposed an interpretable multi-part attention network for brain tumor classification. In detail, the widely used VGG16 was built with a specific interpretable architecture to ensure good enough classification performance for the BRATS 2017 dataset. The model was evaluated in terms of both classification and explainability perspectives. Results demonstrated the model produced accurate tumor classification, and the classification accuracy is on par with some of the best-performing CNN models. Furthermore, the proposed framework is able to provide higher quality explanations for HGG and LGG classification, including global explanation and local explanation.

In detail, global explanation is interpreted as a set of feature patterns the model learns from to classify HGG and LGG. The quality of the feature patterns in terms of their validity and representativity was evaluated by radiologists to see if they were valid evidence for decision aids. Results demonstrated the model learns from the class-representative features of both classes for the classification task, and the HGG feature patterns have higher responses in the contrast-enhancing tumor, necrotic tumor core, and the edematous areas as classification evidence; this agrees with the actual imaging characteristics of HGG. The LGG feature patterns present higher responses on the homogeneous tumor cores and the non-enhancing tumor margins.

Another important advantage of the proposed model is the local explanation it presents for individual predictions. Background areas, such as the ventricles, were found to be activated by the ‘tumor core’ feature patterns of the LGG class. These background patches are not faithful features to the underlying lesion. Therefore, unboxing the reasoning process is necessary; it allows the clinicians and patients to screen out ‘unreliable’ correct predictions.

The local explanation of individual explanations was also evaluated by radiologists to see if it is reliable and acceptable for decision-making support. This form of reliability evaluation and model tuning is not available in the development of “black box” networks or the interpretable models mentioned above. According to the findings, the developed solution provided positive outcomes regarding the brain tumor classification and explanation targeted in this study.

Considering the limitations of the present study, these can be divided into methodological limitations in the construction of the network and limitations in the contextualization of the results.

It is reasonable to suppose that network construction limitations contribute to the lower classification accuracy of the proposed interpretable model compared with the baseline model. This discrepancy could be attributed to the model’s classification inference process, which is greatly influenced by the feature patterns obtained from the randomly generated push dataset. In future work, optimizing the selection of the push dataset may help to improve the classification accuracy of the model. It is also possible that the training data augmentation process could be optimized, as some recent evidence suggests that, even though we used very widely used augmentation methods, the inclusion of image orientations not found in the testing set does not improve the generalizing ability of the model [39].

Regarding interpretation of the results, we did not find other interpretable deep learning methods applied to brain tumor classification based on the same dataset, and we cannot confirm the degree to which the 86% reliability obtained by the model would be considered acceptable by the health workers. Further collaboration with medical practitioners is important for the practical assessment of our model. Considering possible future developments or our work, several possible extensions are clear. The data modalities could be extended to incorporate a greater variety of structural images, such as T1w, T2w, and FLAIR, as well as more targeted sequences, including amide proton transfer [40] and MR spectroscopy [41]. It is also important to consider whether findings in the BraTS2017 dataset carry over into other datasets. For example, many clinical scanners continue to use lower field strengths. Publicly available data sets such as MNIBITE [42] and the recent ReMIND [43] could be leveraged to test IMPA-Net with 1.5-T data.

## 6. Conclusions

An interpretable classification model based on CNN was developed for brain tumor classification to enhance the interpretability and trustworthiness of the model and the health outcomes. The proposed model visualizes the features the model learns and uses for the classification task. It unboxes the reasoning process of individual predictions and explains the outcomes in a human-understandable manner, allowing clinicians and patients to understand and evaluate the reliability of predictions.

In future investigations, alternative datasets encompassing a greater variety of sequences and settings, will be included to improve the classification performance and the generality of the work. Further discussions on the quality of decision aids are also necessary to determine whether they improved decision making and outcomes for patients facing treatment or screening decisions and to explore the applicability of IMPA-Net in other medical imaging tasks.

## Figures and Tables

**Figure 1 diagnostics-14-00997-f001:**
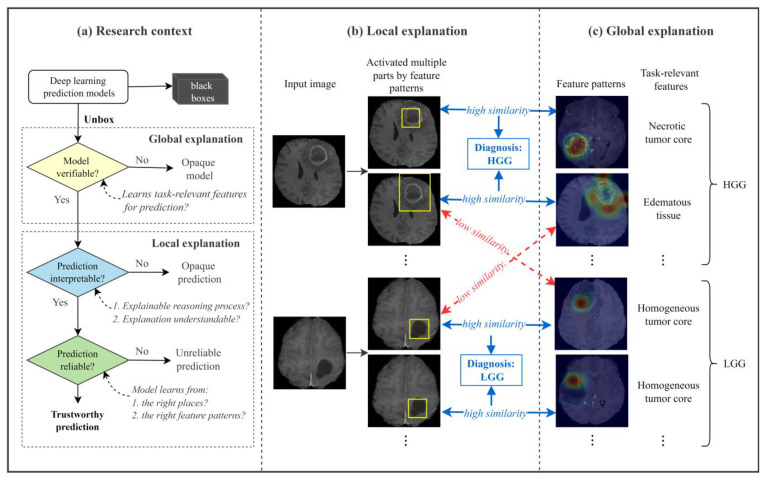
Global and local explanations provided by the proposed IMPA-Net. (**a**) Research context illustrates the importance and basic ideas of global and local explanations for deep learning-based brain tumor classification. It outlines the problems in this research field that the proposed IMPA-Net attempts to address; (**b**) local explanation: given an input image, IMPA-Net compares the activated parts of the input image with the feature patterns and thereby predicts the tumor grade; (**c**) global explanation can be interpreted as the class-representative features the entire model learns to distinguish two classes.

**Figure 2 diagnostics-14-00997-f002:**
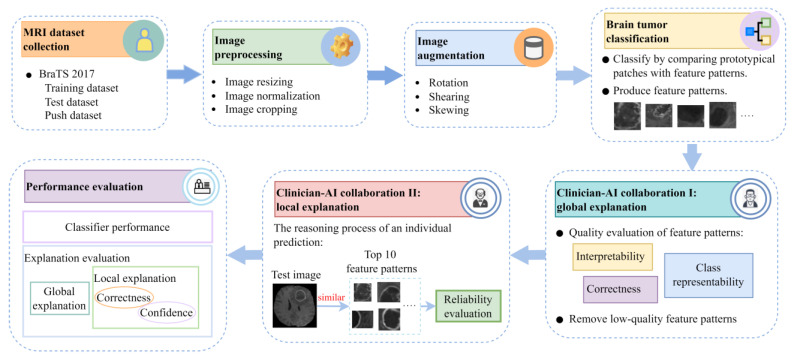
The overall workflow of the development and evaluation of the proposed methodology.

**Figure 3 diagnostics-14-00997-f003:**
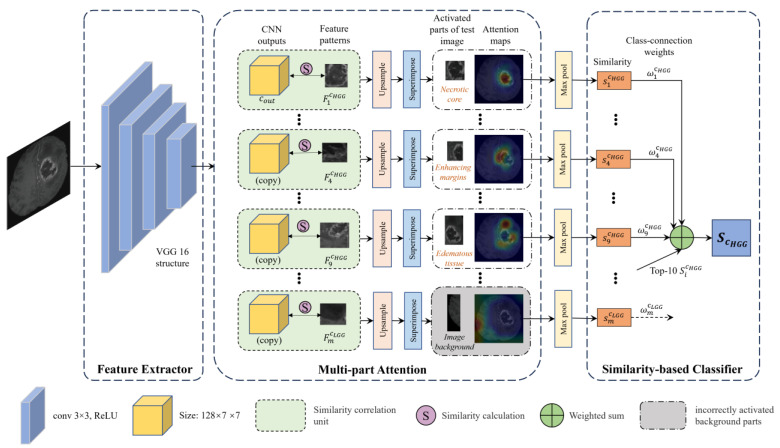
Schematic diagram of the proposed IMPA-Net. It consists of three modules: a feature extractor, a multi-part attention block, and a similarity-based classifier. The feature patterns within the multi-part attention block are learned from the push dataset during the training phase.

**Figure 4 diagnostics-14-00997-f004:**
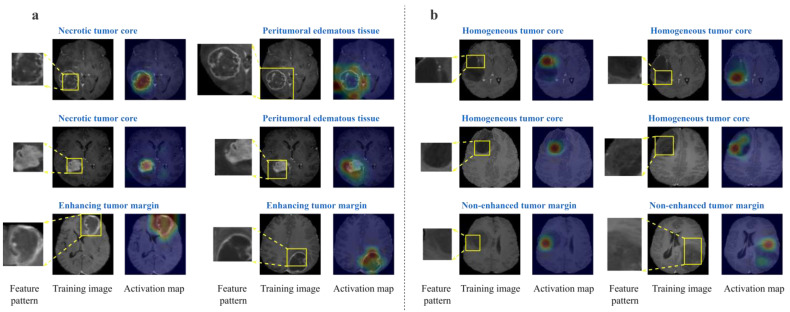
Six learned feature patterns and activation maps of HGG (**a**) and LGG (**b**) selected to represent different clinically relevant discriminative features of each class learned by the model. Training image where feature pattern comes from (feature pattern in box); Activation map (warmer colors indicate higher activation).

**Figure 5 diagnostics-14-00997-f005:**
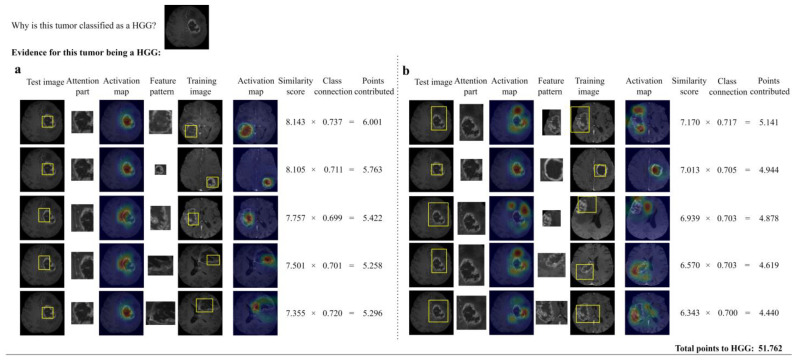
The reasoning process of our network for deciding the grade of a tumor. There are ten rows, split into two groups for ease of presentation: (**a**) top 1~#5th part attention between a patch of the test image and feature pattern, (**b**) #6th~#10th part attention between a patch of the test image and feature pattern. Each row is organized as follows: in the leftmost column a yellow rectangle generated by the proposed model is superimposed on the test image, showing a part that looks like a feature pattern; second column, an enlargement of the part of the test image considered by the model to look similar to the feature pattern (shown in col. 4); third column: activation maps indicating how similar each featured pattern resembles part of the test image, in which warmer color indicates higher responses; fifth column: training images where feature pattern comes from; sixth column: corresponding activation maps. The final columns quantify the result of the comparison. Column 7: similarity score between the localized prototypical part of the test image (col. 2) and the feature pattern (col. 4). Column 8: class connection values generated by the proposed model correspond to the class-connection weight connection between the feature patterns and the logit of class. Column 9: weighted similarity scores between the localized prototypical patches of the test image with top-10 feature patterns.

**Figure 6 diagnostics-14-00997-f006:**
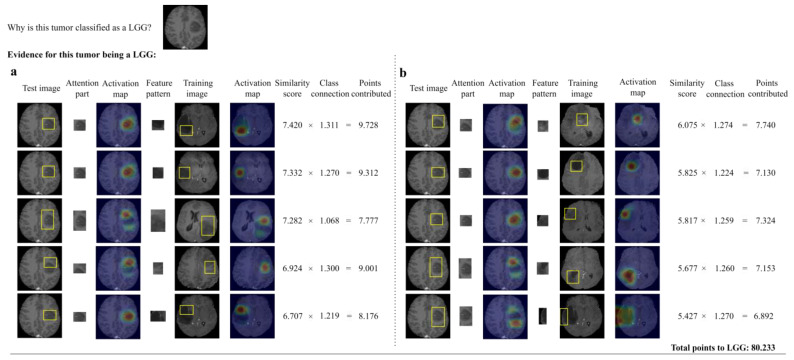
Example output showing the reasoning process of our network in deciding the grade of an LGG tumor, (**a**) top 1~#5th part attention between a patch of the test image and feature pattern, (**b**) #6th~#10th part attention between a patch of the test image and feature pattern.

**Figure 7 diagnostics-14-00997-f007:**
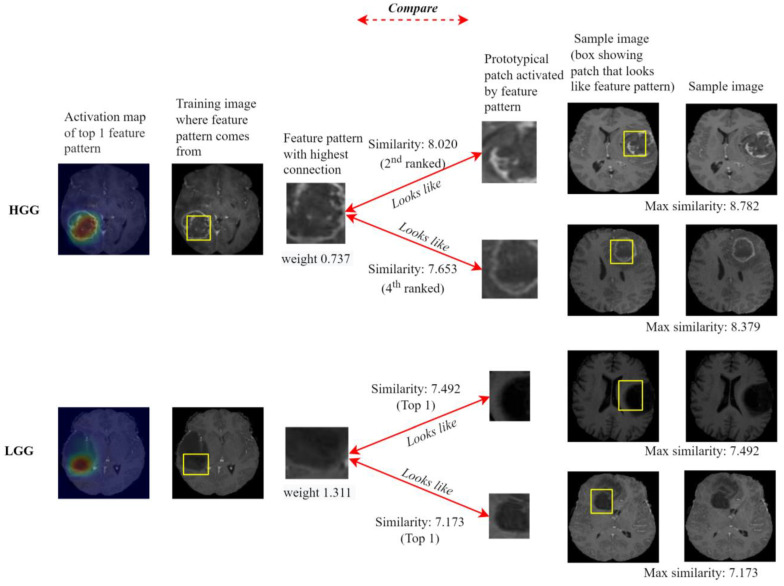
Representability of the feature patterns. The explanations on two input MR images are shown for the feature pattern that has the largest class-connection weight on each class.

**Table 1 diagnostics-14-00997-t001:** Comparison of the classification performance of our interpretable model with the baseline model.

Model	Performance Metrics				
	ACC	PRE	SPE	SEN	F_1_ Score
Baseline model	97.30%	99.18%	98.96%	96.03%	0.9758
Our model before exclusion	85.59%	89.17%	86.46%	84.92%	0.8699
Our model after exclusion	92.12%	94.65%	93.23%	91.27%	0.9293

**Table 2 diagnostics-14-00997-t002:** Summary of the number of feature patterns (top 10) consistent or inconsistent with the actual class of test images among all test cases, unreliable predictions, wrong predictions, and correct predictions (TP and TN predictions), summarized as mean (standard deviation) of the fraction of feature patterns among the top 10 that match or mismatch to the actual class of test images.

Class of Feature Pattern	Class of Test Image							
	All Test Cases		Unreliable Predictions ^1^		Wrong Predictions ^2^		Correct Predictions ^3^	
	HGG	LGG	HGG	LGG	HGG	LGG	HGG	LGG
HGG	9.29 (2.27)	0.90 (2.26)	9.87 (0.41)	1.56 (1.17)	2.09 (1.27)	8.62 (1.26)	9.98 (0.17)	0.34 (0.84)
LGG	0.71 (2.27)	9.10 (2.26)	0.13 (0.41)	8.44 (1.17)	7.91 (1.27)	1.38 (1.26)	0.02 (0.17)	9.66 (0.84)

Note: ^1^ Unreliable predictions are cases among {TP, TN} predictions that are evaluated to be unreliable according to the two identified reliability criteria. ^2^ Wrong predictions are {FP, FN}. ^3^ Correct predictions are {unreliable predictions, reliable predictions}.

**Table 3 diagnostics-14-00997-t003:** The numbers of incorrectly activated background patches by the top 10 feature patterns were given as mean (standard deviation) of the fraction of feature patterns among the top 10 that mismatched the actual class of test images.

Concept of Activated Patch	Class of Test Image							
	All Test Images		Unreliable Predictions		Wrong Predictions		Correct Predictions	
	HGG	LGG	HGG	LGG	HGG	LGG	HGG	LGG
Image background area	0.33 (0.89)	0.46 (1.40)	1.37 (1.34)	1.85 (2.41)	1.46 (1.44)	1.23 (1.83)	0.23 (0.74)	0.40 (1.35)
	0.39 (1.14)		1.61 (1.96)		1.37 (1.57)		0.30 (1.06)	
Brain background area	0.65 (1.55)	0.97 (3.19)	2.61 (2.07)	4.46 (3.67)	2.32 (2.30)	1.00 (0.71)	0.43 (1.28)	0.97 (2.51)
	0.75 (11.96)		3.55 (3.11)		1.83 (1.96)		0.67 (1.93)	

## Data Availability

The first author takes full responsibility for the analyses, interpretation, and conduct of the research. The underlying codes are available from the first author upon reasonable request. The data are publicly available at https://www.med.upenn.edu/sbia/brats2017/data.html (accessed on 10 May 2024).

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
