# Peer review of "IMPA-Net: Interpretable Multi-Part Attention Network for Trustworthy Brain Tumor Classification from MRI"

_diagnostics, 2024, doi:10.3390/diagnostics14100997_

Round 1

Reviewer 1 Report

Comments and Suggestions for Authors

In the paper, an Interpretable Multi-part Attention Network (IMPA-Net) is proposed for brain tumor classification.

The model has a classification accuracy of 92.12%. There are studies in the literature with higher accuracy values for classification. However, the Model both predicts tumor grade and ensures the interpretability of classification results.

1. Why was VGG16 used as a feature extractor? Please detail your reasons.

2. The rotate function was used for data augmenting. It is wrong to do a 20-degree rotation. Because in real life, no doctor or radiologist obtains the patient's brain image by rotating it 20 degrees on the vertical axis. A maximum of 5 degrees for rotation might be reasonable. However, 20 degrees is not suitable. You can discuss this criticism with a radiologist. State your justification in the paper.

3. Briefly discuss the limitations of the model and the literature comparison in the discussion section.

Reviewer 2 Report

Comments and Suggestions for Authors

In the considered manuscript, the authors propose an approach for explaining the outcomes of deep learning-based classification of Brain Tumor from MRI. They devise the global and local explanation components and evaluate their model on the BraTS2017 dataset, comparing the outcome with a baseline.
The topic of explainable AI is an important one, and the authors make a solid contribution in their work. It is also relevant to the scope of Diagnostics, being practical about tumor classification. The technical quality of the paper is very decent, it is well structured and easy to read.
I recommend accepting the manuscript after minor revisions (some suggestions are presented below).

== Dataset ==
Although BraTS2017 is a long-existing dataset, and the authors provide the references ([33],[34]), I would recommend describing it in more detail, so that the reader does not have to consult external sources.

== Discussion ==
* In the Discussion, I would recommend to include the part about generalizing the results to the other MRI scanners, beyond the ones used in BraTS2017 (e.g., 1.5T that is still widely used in medical practice). Also, covering alternative methods such as amid proton transfer or MRS might be of interest.
* I also wonder, would the authors' approach generalize to the classification of the tumors into Grade 2, Grade 3 и Grade 4, which is desirable for the medics.
* I would suggest that the authors devote more of the section to the discussion of the accuracy loss they encountered (compared to the baseline model). If they could detail the reasons for it and outline the ways the drop in the accuracy could be mitigated, this would benefit further research.

== Conclusions ==
The authors claim that "those localized medical features can be understood and interpreted by the users, and thus our framework can help provide global explanations in a human-understandable manner." (actually they repeat this statement twice), but the ground for this is weak - almost no information is disclosed about the evaluation of the explanations with the users/experts (radiologist). The reader also has no idea whether the obtained 86% is a good result. I would suggest that the authors detail this process, and present the relevant data. For instance, what is the degree of the agreement between the two experts? were there any specific problems that caused the -14%?

== Misc ==
* The abbreviations LGG and HGG should be defined before first usage.
* The figures in the manuscript have ALT (mouseover) text in Chinese - should be translated.
* iThenticate indicates a considerable amount of matching text with the following article, but I did not find it in the References. Possibly, the authors might choose to reference it: Singh, G., & Yow, K. C. (2021). These do not look like those: An interpretable deep learning model for image recognition. IEEE Access, 9, 41482-41493.
* at https://www.med.upenn.edu/sbia/brats2017/data.html , they ask to reference two papers by Bakas et al., but I think I found only one in the References.
